# Repeated Oral Administration of Flavan-3-ols Induces Browning in Mice Adipose Tissues through Sympathetic Nerve Activation

**DOI:** 10.3390/nu13124214

**Published:** 2021-11-24

**Authors:** Yuko Ishii, Orie Muta, Tomohiro Teshima, Nayuta Hirasima, Minayu Odaka, Taiki Fushimi, Yasuyuki Fujii, Naomi Osakabe

**Affiliations:** 1Functional Control Systems, Graduate School of Engineering and Science, Shibaura Institute of Technology, 307 Fukasaku, Minumaku, Saitama 337-8570, Japan; mf19009@shibaura-it.ac.jp (Y.I.); mf21121@shibaura-it.ac.jp (O.M.); mf19052@shibaura-it.ac.jp (T.T.); mf19064@shibaura-it.ac.jp (N.H.); nb21110@shibaura-it.ac.jp (T.F.); nb19105@shibaura-it.ac.jp (Y.F.); 2Department of Bio-Science and Engineering, Shibaura Institute of Technology, 307 Fukasaku, Minumaku, Saitama 337-8570, Japan; bn18015@shibaura-it.ac.jp

**Keywords:** flavan-3-ols, adipose, browning, catecholamine, sympathetic nerve

## Abstract

We previously found increases in uncoupling protein (Ucp)-1 transcription in brown adipose tissue (BAT) of mice following a single oral dose of flavan 3-ol (FL)s, a fraction of catechins and procyanidins. It was confirmed that these changes were totally reduced by co-treatment of adrenaline blockers. According to these previous results, FLs possibly activate sympathetic nervous system (SNS). In this study, we confirmed the marked increase in urinary catecholamine (CA) s projecting SNS activity following a single dose of 50 mg/kg FLs. In addition, we examined the impact of the repeated administration of 50 mg/kg FLs for 14 days on adipose tissues in mice. In BAT, FLs tended to increase the level of Ucp-1 along with significant increase of thermogenic transcriptome factors expressions, such as *peroxisome proliferator-activated receptor γ coactivator (PGC)-1α* and *PR domain-containing (PRDM)1*. Expression of browning markers, *CD137* and *transmembrane protein (TMEM) 26*, in addition to *PGC-1α* were increased in epididymal adipose (eWAT) by FLs. A multilocular morphology with cell size reduction was shown in the inguinal adipose (iWAT), together with increasing the level of Ucp-1 by FLs. These results exert that FLs induce browning in adipose, and this change is possibly produced by the activation of the SNS.

## 1. Introduction

Flavan-3-ol (FL)s, a fraction of B-type oligomer procyanidins (Figure 1b) slightly containing catechins (Figure 1c), are enriched in cocoa [1,2], apple [3,4], grape seeds [5,6], and red wine [7,8]. The ingestion of FL-rich foods could have a significant potential for managing cardiovascular health [9,10,11,12]. Many intervention studies have suggested that the intake of FLs results in beneficial alterations in the metabolism, such as a significant decrease in plasma LDL with an increase in HDL [13,14,15] and a rise in glucose tolerance [16,17,18].

A part of the catechins is absorbed in the gastrointestinal tract and, subsequently, are metabolized in intestinal epithelial cells or the liver. Therefore, unchanged forms of them are nearly absent in blood or tissues [19]. In addition, B-type oligomer procyanidins are rarely absorbed from the gut into the blood [20,21,22]. Almost all B-type procyanidins ingested from foods move into the colon, and a part of them are degraded by the microbiome. It has been reported that the gut microbiome and their metabolites are altered by ingestion of FLs over a long period. These conversions possibly contribute to improvements in metabolism. [23,24,25,26,27].

In contrast, acute metabolic changes, such as improvements in glucose and insulin tolerance, were shown after a single intake of FL-rich food [28,29]. These previous results point out the necessity of discussions from both the acute and chronic sides of the physiological alterations induced by FLs ingestion.

Catecholamine (CA)s, which are secreted from the end of the sympathetic nerve to effector tissues, are known to induce large metabolic alterations through the activation of adrenaline receptors (AR), including β3AR [30]. Activation of β3 AR-expressed brown adipose tissue (BAT) induced thermogenesis via the activation of mitochondrial uncoupling protein 1 (Ucp-1) dissipating energy as heat. It has been well-investigated that SNS activation, induced by cold stress or treatment of adrenaline agonists, enhances non-shivering thermogenesis through the activation of Ucp-1 [31].

We previously found that a significant increases in energy expenditure and *UCP*-1 mRNA expression in BAT a few hours after a single oral dose of FLs in mice [32]. We also confirmed that these changes were completely reduced by co-treatment with FLs and β3 adrenaline blockers [33]. In addition, our previous results suggested that a single oral dose of FLs induced stress response. In the response to stress, sympathetic hyperactivity occurs together with hypothalamic–pituitary–adrenal (HPA) axis activation [34]. Therefore, our previous results in these metabolic changes suggest that FLs possibly induce sympathetic nervous system (SNS) activation.

Recently, it was reported that the activation of SNS induces the browning of white adipose tissue (WAT), such as the conversion of WAT into beige adipocytes or the promotion of differentiation from mesenchymal stem cells to beige adipose [35]. Beige adipocytes, as brown-like adipocytes, were found after SNS activation with cold exposure [36]. Beige adipocytes have been characterized by a multilocular morphology and the expression of Ucp-1.

Besides CAs, such as adrenaline (AD) and noradrenalin (ND) are also secreted from the adrenal medulla to blood following the activation of the SNS. Plasma AD and ND comprising 60–70% sulfated conjugated metabolites are excreted into the urine [37]. In clinical analysis, the correlation between sympathetic nerve activation and plasma or urine levels is controversial. In contrast, the correlation between them was verified in various rodent stress models [38,39,40]. As described before, sympathetic hyperactivity following a single oral dose of FLs may occur as a stress response [41]. Therefore, we employed the determination of AD and ND levels in urine to estimate the change in SNS activity.

In the present study, we first determined the AD and ND concentration in urine over 24 h, following a single oral dose of FLs, to estimate the activity of sympathetic nerves. Next, we examined the effects of the repeated oral dose of FLs on mouse adipose tissues, such as BAT, epididymal (eWAT), and inguinal white adipose (iWAT), by the measurement of thermogenic-related protein and gene expressions. In addition, we carried out histological observation to evaluate browning in iWAT.

## 2. Materials and Methods

### 2.1. Materials

FLs from cocoa were prepared according to Natsume et al. [42]. The FLs contained 4.56% (+)-catechin, 6.43% (−)-epicatechin, 3.93% procyanidin B2, 2.36% procyanidin C1, and 1.45% cinnamtannin A2 (Figure 1a–c). To determine a reference value, we also ascertained the polyphenol concentration in this fraction using the Prussian blue method, and it showed a value of 72.3%.

### 2.2. Animals and Diets

The study was approved by the Animal Care and Use Committee of the Shibaura Institute of Technology (Permit Number: AEA19016). We used C57BL/6J 13-week-old mice, weighing 21–26 g, obtained from Charles River Laboratories Japan, Inc. (Tokyo, Japan). In the acclimation period, four mice were placed in a cage and kept in a temperature-regulated room (23–25 °C) with controlled lighting (12/12-h light/dark cycles) and freely accessible water and diet. Basal diet (MF^®^) was obtained from Oriental Yeast Co., Ltd. (Tokyo, Japan).

### 2.3. Impact of a Single Dose of FLs on the Excretion of CAs in Urine

After being fed a basal diet for 14 days, the mice were divided randomly into two treatment groups as follows: vehicle (3% Tween 80 in distilled water) (*n* = 8); 50 mg/kg FLs (*n* = 8). Animals were placed individually in a cage for urine collection (Figure 1e) and were able to access food and water freely. Following a 48-h habituation period, 24-h urine, both pre- and post-oral administration of vehicle or FLs, was collected using 20 μL of a 2.5 mol/L HCl-containing tube, as shown in Figure 1d. The oral administration of the vehicle or FLs was carried out between 10:00 and 11:00 AM.

### 2.4. Analyses of CAs in Urine

We determined the concentration of urinary CAs, such as noradrenaline (NA), adrenaline (AD) and their metabolites, by treatment with the enzyme (sulfatase from Helix pomatia Type H-2, Sigma–Aldrich, St. Louis, MO, USA) according to the previous report [43]. Briefly, the urine was heated for 10 min following incubation with 500 U/mL of enzyme at 37 °C for one hour. After addition of isoprenaline (Sigma-Aldrich, St. Louis, MO, USA) as the internal standard, CAs were purified using a Monospin PBA (GL sciences, Tokyo Japan). The HPLC system (Prominance HPLC System Shimazu Corporation, Kyoto, Japan) consisted of a quaternary pump with a vacuum degasser, thermostatted column compartment, autosampler, and equipped with an electrochemical detector (ECD 700 S, Eicom Corporation, Kyoto, Japan). A reverse-phase column (Inertsil ODS-4; 250 × 3.0 mm ID, 5 µm, GL Sciences, Tokyo, Japan) was used, and the column temperature was maintained at 35 °C. The HPLC mobile phase was 24 mM acetate–citrate buffer (pH 3.5)-CH_3_CN (100/14.1, *v/v*). The mobile phase flow rate was 0.3 mL/min, and the injection volume was 20 μL. The eluents were detected and analyzed at 500 mV. Excretion of CAs were expressed as a ratio, with urinary creatinine concentration measured using a LabAssay Creatinine (FUJIFILM Wako Pure Chemical Corporation, Tokyo, Japan).

### 2.5. Impact of the Repeated Oral Administration of FLs on Mouse Adipose Tissues

After being fed a basal diet for 14 days, mice were divided randomly into two treatment groups as follows: vehicle (3% Tween 80 in distilled water, *n* = 11); 50 mg/kg FLs (*n* = 11). Gavage administration of vehicle or 50 mg/kg bw FLs, for 14 days for mice in each group were, is shown in Figure 1f. At the end of the treatment period, to avoid suffering and the influence of anesthesia on blood CA dynamics, all the animals were sacrificed via decapitation by skilled researchers and according to the experimental procedures.

BAT, eWAT, and iWAT were collected from each mouse by dissection. BAT and eWAT were snap-frozen in liquid nitrogen and stored at −80 °C until analysis. Six out of 11 of iWAT were also stored as described above. The other 5 out of 11 iWAT were used as frozen sections for histological observation.

### 2.6. Quantitative RT-PCR Analysis

Total RNA was prepared from BAT and iWAT using TRIzol Reagent (Life Technologies, Carlsbad, CA, USA) and from eWAT using QIAzol (Qiagen, Hilden, Germany) according to the manufacturers’ instructions. Briefly, 10 μg of total RNA was reverse-transcribed in a 20-μL reaction volume with High-Capacity cDNA Reverse Transcription Kits (Life Technologies, Carlsbad, CA, USA). Real-time reverse-transcription PCR was performed using 50 ng of total cDNA in the StepOne^™^ Real-Time PCR System (Life Technologies, Carlsbad, CA, USA). The primer and probe sequences were selected using a Taqman^™^ Gene Expression Assay (Life Technologies, Carlsbad, CA, USA) and the following genes: glyceraldehyde-3-phosphate dehydrogenase (*GAPDH*) (Mm99999915_g1), *β-actin* (Mm02619580_g1), *UCP-1* (Mm01244861_m1), *peroxisome proliferator-activated receptor γ coactivator* (*PGC)-1α* (Mm01208835_m1), *PR domain-containing* (*PRDM)16* (Mm00712556_m1), *CD137* (Mm00441899_m1), *transmembrane protein (TMEM)26* (Mm01173641_m1), and *T-Box Transcription Factor (TBX)-1* (Mm00448949) from Life Technologies; *GAPDH* and *β-actin* were used as internal controls. The buffer used in the systems was THUNDER BIRD Prove qPCR Mix (TOYOBO, Tokyo, Japan). The PCR cycling conditions were 95 °C for 1 min, followed by 40 cycles at 95 °C for 15 s and 60 °C for 1 min.

### 2.7. Western Blotting

Tissues were homogenized in a microtube with lysis buffer (CelLyticTM MT Cell Lysis Reagent; Sigma-Aldrich, Japan) containing a protease inhibitor (Sigma–Aldrich, Japan) and 0.2% *w*/*v* SDS. Protein concentration was measured using the Bradford method. Protein (20 µg) was separated by SDS-PAGE using a 10–20% Bis-Tris gel and transferred onto a polyvinylidene difluoride membrane (Life Technologies, CA, USA). The membrane was blocked with membrane-blocking reagent (GE Healthcare, Buckinghamshire, UK) for one hour. After blocking, the membrane was incubated with a rabbit polyclonal primary antibody against α-tubulin (1:800; GR3224374-1, Abcam, Cambridge, UK) and an antibody against Ucp-1 (1:1000; GR286375-2, Abcam). After the primary antibody reaction, the membrane was incubated with appropriate horseradish peroxidase-conjugated secondary antibodies (1:100,000, Anti-Rabbit IgG, HRP-Linked Whole Ab Sheep, NA931, GE healthcare, Buckinghamshire, UK) for one hour. Immunoreactivity was detected by chemiluminescence using the ECL Select Western Blotting Reagent (GE Healthcare, Buckinghamshire, UK). Fluorescence band images were analyzed using Just TLC (SWEDAY, Larkgatan, Sweden) analysis software. The ratio of Ucp-1 to α-tubulin for each animal was calculated.

### 2.8. Histological Analysis of iWAT

iWAT was fixed using a Softmount (192-10301, FUJIFILM Wako Pure Chemical Corporation, Tokyo, Japan). Ten-micrometer-thick histological sections were cut and stained with hematoxylin and eosin (H&E). All observations were performed with an uplit microscope (CX41LF, OLYMPUS CORPRATION). Histological observation was carried out on H-and-E-stained slides at a magnification of 40× using a Camera Control Pro 2, Nikon.

### 2.9. Data Analysis and Statistical Methods

The data are expressed as means and standard deviations. Statistical analyses were performed using two-way ANOVA followed by the post hoc Dunnett’s test or the Student’s *t*-test (BellCurve for Excel, Social Survey Research Information Co., Ltd.). *p* < 0.05 was considered significant, and *p* < 0.1 was considered to tend toward significance.

## 3. Results

### 3.1. Twenty-Four Hours Urinary CAs Both Pre-and Post-Oral Administration of FL in Mice

The levels of excreted CAs and their metabolites in 24-h urine pre and post a single dose of vehicle or 50 mg/kg FLs are shown in Figure 2. The excretion amount of NA during 24 h was almost similar in both the pre- and post-vehicle groups and the pre-FLs group; in contrast, a single oral dose of FLs significantly increased NA excretion in urine (Figure 2a). Nearly similar results as from NA were obtained in AD urine levels, as shown in Figure 2b. The total CAs, which is the sum of NA and AD, significantly increased following a single oral administration of FLs (Figure 2c).

### 3.2. Impact of Repeated Oral Doses of FLs Adipose Tissues in Mice

The levels of thermogenic-related protein (Ucp-1) and gene expressions (*UCP-1*, *PGC-1α, PRDM16*) in BAT are shown in Figure 3. Ucp-1 tended to increase in BAT by the repeated administration of FLs (Figure 3a). FLs treatment significantly increased the mRNA expression of *UCP-1*, *PGC-1α,* and *PRDM16* in BAT as shown in Figure 3b.

Thermogenic-related protein and genes levels, and the expressions of beige markers, such as *CD137, TMEM26,* and *TBX-1,* in eWAT are shown in Figure 4. There was no difference in the levels of Ucp-1 between the experimental groups (Figure 4a). The mRNA expression of *PGC-1α* was significantly increased. The expression of *CD137* was significant increased, *TMEM26* was slightly increased, but not *TBX-1*, following repeated gavage treatment of FLs (Figure 4b). The results of histochemical observations in iWAT, the level of thermogenic-related protein and genes, and beige markers are shown in Figure 5. In multiple sections in all of iWAT in the FLs group, we observed the multilocular morphology along with a reduction in cell size, which is shown in the right panel of Figure 5a. Yet, these changes were not shown in the vehicle group. The level of Ucp-1 increased significantly in iWAT of the FLs group compared with the vehicle group (Figure 5b). *PRDM16* mRNA expression was significantly increased in the FLs group (Figure 5c).

## 4. Discussion

The activation of BAT and the formation of brown-like adipocytes called beige adipocytes, within WAT, has been the focus of much attention as a therapeutic target for obesity and its complications [44]. These adipose tissues have attracted special interest because of their possible ability to differentiate themselves from white adipocytes and their ability to dissipate energy [45]. It was well investigated that the browning of adipose tissue is induced by cold exposure [46], exercise [47], and caloric restriction [48]. The browning is produced by the activation of the SNS [49]. SNS-innervating adipose, such as BAT or beige adipose, plays a key role in promoting non-shivering thermogenesis.

Recently, it has been reported that dietary polyphenols, such as catechin [50] and resveratrol [51], were reported to activate BAT and possibly induce thermogenesis. Choo et al. also reported that a body fat-suppressive effect induced by green tea catechin was inhibited by the supplementation of a β blocker [50]. We previously found that a single oral dose of FLs induced enhancement of energy expenditure along with increased the mRNA expression of *UCP*-1 in BAT [52]. These changes were totally reduced by co-administration of a specific β3 blocker, SR52930 33. A single oral administration of FLs also increased blood flow in skeletal muscle [52] and this alteration may be exerted through the activation of β1 and β2 adrenaline receptors [53].

In addition, we observed that a single oral dose of FLs induced a stress response with the upregulation of corticotropin-releasing hormone (CRH) in paraventricular hypothalamic nucleus and increase of plasma corticosterone [41]. The stressors result in rapid activation of the HPA axis and SNS, consequently, NA, AD, and corticosterone are secreted from the adrenal gland to the blood, while NE is secreted from the end of the sympathetic nerve to the effector, including BAT [34]. According to these previous results, it is suggested that oral administration of FLs activates SNS and, consequently, causes various physiological alterations. Although the involvement of subclasses of β-adrenergic receptor in adipose browning is not fully understood, it is considered that repeated administration of FLs may producing browning via catecholamines and its receptors, according to the results of a previous and this experiment.

Accordingly, we first determined the amount of CAs excreted into urine following a single oral dose of FLs. A significant increase in the amount of NA and AD excretion of 24-h urine was observed following administration of FLs (Figure 2). Determination of urinary CAs has been used to assess SNS activity induced by stressors in mammals [54]. Especially, the correlation between the activity of SNS and the level of plasma or urine CAs was verified in rodent stress models, such as social defeat [38], hypothermia [39], or PTSD models [40]. Oral administration of FLs likely activate SNS because they are considered stressors in rodent models. Therefore, increasing CAs and their metabolites, following a single oral dose of FLs, was considered the result of the activation of SNS induced by FLs.

Next, we examined the impact of the repeated oral doses of FLs on adipose in mice. The results showed that the level of Ucp-1 tended to increase and its mRNA was significantly upregulated in the BAT of the FLs group (Figure 3a,b). The activation of the β3 receptor in BAT or beige adipose induces the activation of the cAMP/PKA pathway, consequently upregulating PGC1α through the activation of p38 mitogen-activated protein kinase (MAPK) [55]. PGC1α is known to be widely expressed in BAT, to upregulate *UCP-1*, and also to increase the number of mitochondria with oxidative capacity, therefore being essential for thermogenesis [56]. It has also been reported that PRDM16 is a transcription factor related to thermogenesis and PGC1α. They work together to induce the development of the BAT phenotype by β3 activation [57].

FLs upregulated these two transcription factors and *UCP-1* in BAT, as shown in Figure 3b. These results suggest that the thermogenic formation of BAT through the transcription of *PRDM16* and *PGC1α* is possibly induced by repeated oral doses of FLs.

In response to SNS activation, beige adipose, characterized by Ucp-1 expression and mitochondrial generation, is reported to develop in WAT [58]. Beige adipose has a thermogenetic ability through the activation of Ucp-1, similarly to BAT. Additionally, it has been suggested that human BAT is closer to the beige adipose tissue of mice than the BAT of mice [59]. In mice, a distinct sub-population of WAT-resident progenitors, which express the markers such as CD137, TMEM26 and TBX -1, show a greater ability to differentiate into beige cells [59].

It was reported that mild SNS activation recruits different precursor cells, such as Myh11^+^ or SMA^+^ vascular smooth muscle cells [60], and Ebf2^+^ or Pdgfrα^+^ adipogenic cells [61] into iWAT, which can be differentiated into beige adipocytes. If SNS inactivation occurs, SNS-induced beige adipocytes lose Ucp-1 expression [62]. De-activated beige cells have a white adipose-like morphology but can be re-activated by an additional bout of β-adrenergic signaling [63]. It was also reported that the differentiation of Pdgfrα^+^ adipogenic cells in eWAT was regulated by SNS activation, resulting in the repeated differentiation from white to beige or beige to white [64]. In addition, iWAT is reported to be highly susceptible to browning, even with mild SNS activation.; whereas the eWAT of male mice is reported to have quite a resistance to browning [63].

Comparing the expression of these beige markers and pathological changes characterized with beige adipose, the response to FLs was different between two types of WAT, visceral eWAT and subcutaneous iWAT. Repeated 14-day gavage treatment with 50 mg/kg FLs significantly increased Ucp-1 levels, along with a multilocular morphology characterized as beige adipose in iWAT (Figure 5). In eWAT, in contrast, the upregulation of browning markers, such as *CD137* and *TMEM26*, was observed, but there was no change in Ucp-1 level following the treatment with FLs (Figure 4).

In the previous study, the expression of beige markers and pathological change under sympathetic hyperactivity were verified in mice WAT [65]. Following cold stress (at 4 °C for 30 days), UCP-1 mRNA was significantly increased in both eWAT and iWAT, while the mRNA levels of browning markers, such as *CD137, TMEM 26,* or *TBX-1* were decreased. These results have shown that even eWAT, which is resistant to SNS, completely differentiates into beige adipocytes, which upregulated *UCP-1* and reduced beige markers when the sympathetic hyperactivity continues for a long period. Under the conditions of our study, if the activation of the sympathetic nerve is milder than 4 °C for 30 days, iWAT, which is easy to differentiate by SNS, may differentiate into beige. In contrast and under the same conditions, eWAT may be still in the process of differentiating to beige adipose expressing these markers.

In the present study, we observed the browning of adipose by the repeated treatment of FLs, but their mechanism of action, including their induction of the activation of the SNS via a stress response, is still unknown.

## 5. Conclusions

In conclusion, the results of this study suggest that oral administration of FLs activated the SNS and, consequently, induced the activation of BAT, promoting browning in WAT. It was suggested that the effect of FLs was more sensitive in subcutaneous iWAT than in visceral eWAT. The mechanisms for the activation of SNS via a stress response induced by the oral administration of FLs or FL-rich foods remain unclear; further elucidation through additional study is needed.

## Figures and Tables

**Figure 1 nutrients-13-04214-f001:**
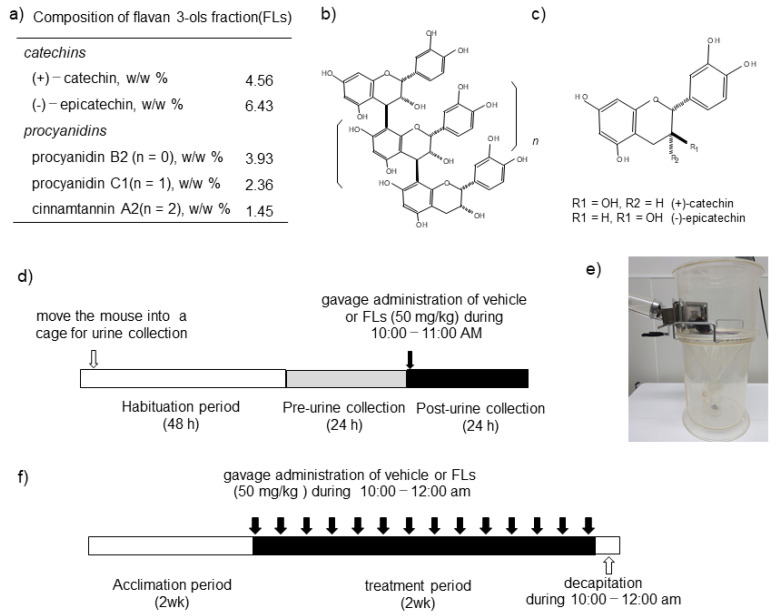
Composition of flavan-3-ols fraction (FLs) and scheme of experimental procedures. Composition of flavan-3-ols fraction (FLs) (**a**), the structure of B-type procyanidins (**b**), the structure of catechins (**c**), the scheme for the experimental procedure to determine urinary catecholamines (CAs) following a single dose of FLs (**d**), photo of the cage for urine collection (**e**), the scheme for the experimental procedure to examine the impact of repeated administration of FLs on adipose tissue in mice (**f**).

**Figure 2 nutrients-13-04214-f002:**
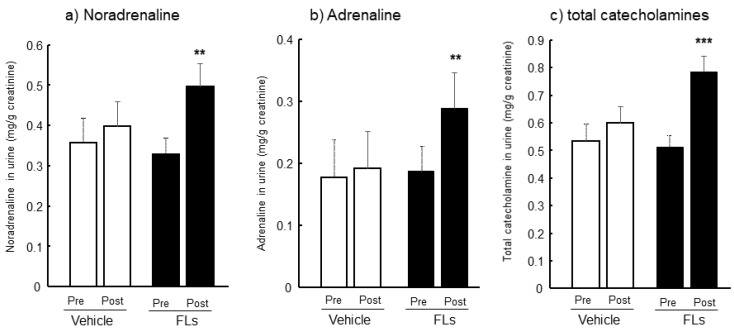
Twenty-four-hours urinary catecholamine (CA)s both pre-and post-oral administration of vehicle or 50 mg/kg flavan-3-ol (FL) s in mice. Noradrenaline (NA, (**a**), adrenaline (AD, (**b**), and total CAs (**c**). The values represent the mean ± standard deviation (each group, *n* = 8). ** *p* < 0.01, *** *p* < 0.001 (two-way ANOVA, followed by the post -Dunnett’s test).

**Figure 3 nutrients-13-04214-f003:**
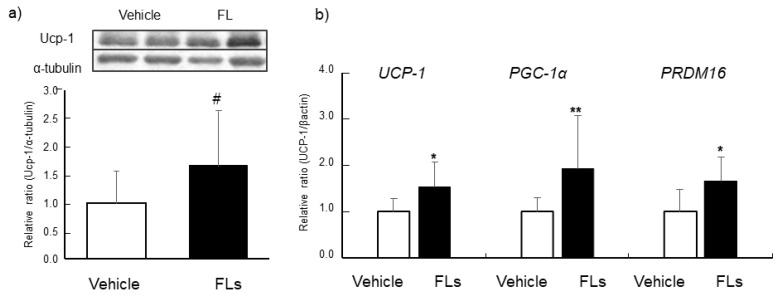
The impact of repeated oral administration of 50 mg/kg flavan-3-ol (FL) s on mice brown adipose tissue (BAT). Ucp-1 levels determined by Western blotting (**a**), mRNA expressions of UCP-1, PGC-1α and PRDM16 (**b**). The values represent the mean ± standard deviation (each group, *n* = 11). # *p* < 0.1, * *p* < 0.05, ** *p* < 0.01 (Student’s t-test).

**Figure 4 nutrients-13-04214-f004:**
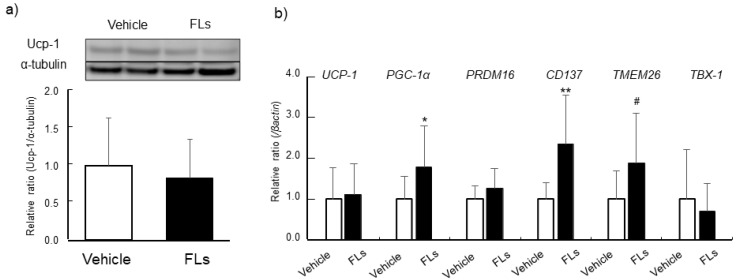
The impact of repeated oral administration of 50 mg/kg flavan 3-ols (FL)s on mice epididymal adipose (eWAT). Ucp-1 levels determined by western blotting (**a**), mRNA expressions of *UCP-1, PGC-1α, PRDM16, CD137, TMEM26* and *TBX-1* (**b**). The values represent the mean ± standard deviation (each group, *n* = 11). # *p* < 0.1, * *p* < 0.05, ** *p* < 0.01 (Student’s *t*-test).

**Figure 5 nutrients-13-04214-f005:**
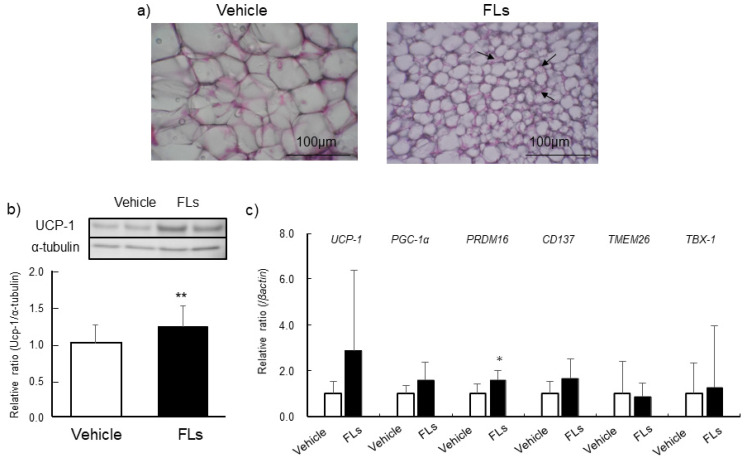
The impact of repeated oral administration of 50 mg/kg flavan 3-ols (FL)s on mice inguinal adipose (iWAT). Histochemical observation of iWAT(**a**) arrows point to multilocular morphology, each group, *n* = 5), Ucp-1 levels determined by western blotting (**b**), mRNA expressions of *UCP-1, PGC-1α, PRDM16, CD137, TMEM26 and TBX-1* (**c**). The values represent the mean ± standard deviation (each group, *n* = 6). * *p* < 0.05, ** *p* < 0.01 (Student’s *t*-test).

## Data Availability

Not applicable.

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
