# Peer review of "Repeated Oral Administration of Flavan-3-ols Induces Browning in Mice Adipose Tissues through Sympathetic Nerve Activation"

_nutrients, 2021, doi:10.3390/nu13124214_

Round 1

Reviewer 1 Report

Some parts in the introduction are not very well connected and some references are not updated.

(e) "metabolic cage" is confusing. It seems you measured O2 consumption but instead is just a way to collect urine 

(f) in missing in the explanation of Fig.1 Line 38

-Statistic analysis in the method is not explained properly. Which program have you used ? Mention the n of samples used for the analysis 

-The results are very synthetics and not extensively explained 

Author Response

Manuscript No.: Nutients-1401797

Title: Repeated oral administration of flavan-3-ols induces browning in mice adipose tissues through sympathetic nerve activation

Corresponding Author: Prof. Naomi Osakabe

Submit Date: 7 Nov 2021

RESPONSE TO REVIEWER 1: We wish to express our appreciation to Reviewer 1 for their insightful comments, which have helped us improve our paper significantly.

Comment 1

 Some parts in the introduction are not very well connected and some references are not updated.

Response:We thank Reviewer 1 for this pertinent comment. According to the comments of Reviewer 1, we have made major revisions. Please see the revised version.

Comment ï¼’

"metabolic cage" is confusing. It seems you measured O2 consumption but instead is just a way to collect urine

Response:We revised from “metabolic cage” to “cage for urine collection”(p7L112 and Figure 1).

Comment 3

(f) in missing in the explanation of Fig.1 Line 38 

Response: We added Fig.1(f) in Figure Legend.

Comment 4

-Statistic analysis in the method is not explained properly. Which program have you used ? Mention the n of samples used for the analysis 

Response: We used BellCurve for Excel(Social Survey Research Information Co., Ltd.) .We added that in “ 2.9. Data analysis and statistical methods”(p12L190) and n of samples in Legend of Figure 5.

Comment 5

-The results are very synthetics and not extensively explained 

Response:According to the comment of Reviewer 1, we have made major revisions. Please see the revised version.

Reviewer 2 Report

Previously, this research group reported that a single dose flavan-3-ols (FL) administration increased energy expenditures in mouse in vivo. In this current study Ishii et al further investigated the effect of repeated administration of FL for 2 weeks on adipose tissues. While studying acute vs. chronic effect of FL in vivo may be valuable, the lack of novelty raises concerns impact of this study.

My detailed suggestions follow below

Major points

  1. Authors in their previous studies showed that ADRB inhibitor blocked FL (single dose)-stimulated energy expenditures. Does ADRB blocker also attenuated the effect of repeated application of FL? Which ADRB, for example b1, b2 or b3 receptors, are mainly involved with FL-mediated effect? In other words FL effect restricted to mature adipocytes or also to preadipocytes? These results better to be provided.
  2. It may be more interesting to understand how FL can be converted or FL stimulates catecholamine synthtesis? If authors don’t have data, this should be speculated in the discussion.
  3. Are the pictures of Figure 3a, 4a, 5a tissues of FL treated? Where are the control tissues? And there is no description or explanation about these figures in the text at all.
  4. Although authors quantified the protein levels of UCP1, there seems to have very mild difference of UCP1 expression between control and FL applied tissues. Better to show more westernblot results to convince the readers.
  5. In line 66~68, I thought that it is controversial about the correlation between CAs in the urine/plasma and SNS activity. Please check multiple references to confirm it.
  6. Line 246~248, Does PRDM16 and PGC1a mediated BAT development through b3 receptor?
  7. Figure 5b: Are the two pictures representative ones? Those may be too exaggerated ones?
  8. Line 277, there is no results showing that FL activated SNS, please edit this text in a prudent way.
  9. Catecholamine downstream signaling pathway should be analyzed in BAT, eWAT, and iWAT. (PKA signaling, pHSL etc.)

Minor points.

  1. The order of Figures should be fixed.
  2. In the method section, most reagents were well described, however information for some reagents, for example secondary antibody for westernblot, are missing (company, city, country). Please review through the text
  3. What is the meaning of upregulation of TMEM26, CD137 (pre brown adipocyte markers?) and PGC1a in eWAT but not in iWAT by FL?
  4. There are citation errors and text, for example references 30 does not have any data mentioned in the text

Author Response

Manuscript No.: Nutients-1401797

Title: Repeated oral administration of flavan-3-ols induces browning in mice adipose tissues through sympathetic nerve activation

Corresponding Author: Prof. Naomi Osakabe

Submit Date: 7 Nov 2021

RESPONSE TO REVIEWER 2: We wish to express our appreciation to Reviewer 2 for their insightful comments, which have helped us improve our paper significantly.

Comment 1

Authors in their previous studies showed that ADRB inhibitor blocked FL (single dose)-stimulated energy expenditures. Does ADRB blocker also attenuated the effect of repeated application of FL? Which ADRB, for example b1, b2 or b3 receptors, are mainly involved with FL-mediated effect? In other words FL effect restricted to mature adipocytes or also to preadipocytes? These results better to be provided.

Response:We thank reviewer 2’s comment. We did not carry out repeated co-treatment of FLs and ADRB, because it was known repeated doses of ADRB cause serious side effects in heart. In our previous results, a single oral dose of FLs transiently rose heart rate and blood pressure via SNS activation in rats, and these changes were reduced by β1 and β2 ADRM carvedilol (Free Radic Biol Med. 2016, 99, 584-592). Therefore, we consider the effect of FLs is not specific for β3 adrenaline receptor. In this paper, we did not mention these results in circulation of rats, because we think that they might confuse readers.

Comment 2

It may be more interesting to understand how FL can be converted or FL stimulates catecholamine synthesis? If authors don’t have data, this should be speculated in the discussion.

Response:Therefore, increase of urinary CAs projecting  sympathetic activation may be induced by stress response by FLs . According to the comment of Reviewer 2, we added the statement as follows

Introduction (p5L70)

In addition, our previous results suggested that a single oral dose of FLs induced stress response. In the stress response, sympathetic hyperactivity occurs together with hypothalamic-pituitary-adrenal (HPA) axis activation34. Therefore, our previous results in these metabolic changes suggest that FLs possibly induce sympathetic nervous system (SNS) activation.

Discussion (p15L238)

In addition, we observed that a single oral dose of FLs induced stress response with upregulation of corticotropin-releasing hormone (CRH) in paraventricular hypothalamic nucleus and increase of plasma corticosterone 41. The stressors result in rapid activation of the HPA axis and SNS, consequently, NA, AD, and corticosterone are secreted from the adrenal gland to the blood, while NE is secreted from the end of the sympathetic nerve to the effector including BAT 34. According to these previous results, it was suggested that oral administration of polyphenols activates SNS, resulting in thermogenesis through the β3 adrenergic receptor expressed in BAT.

.

Comment 3

Are the pictures of Figure 3a, 4a, 5a tissues of FL treated? Where are the control tissues? And there is no description or explanation about these figures in the text at all.

Response:Photo 3a, 4a, 5a is just a photo of the excised part of each tissue. We will delete it if there is no particular need, but what should we do?

Comment 4

Although authors quantified the protein levels of UCP1, there seems to have very mild difference of UCP1 expression between control and FL applied tissues. Better to show more western blot results to convince the readers.

→We revised Figures 3,4,and 5.

Comment 5

In line 66~68, I thought that it is controversial about the correlation between CAs in the urine/plasma and SNS activity. Please check multiple references to confirm it.

Response:We thank Reviewer 2’s comment. We understood the controversy about the correlation between CAs and SNS activity. In contrast, the correlation between blood or urinary CAs and sympathetic nerve activity was verified by several rodent stress models. Additionally, we added that we observed FLs caused a stress response as described above. According to the comment of Reviewer 2, we also added the statements as follows.

Introduction(p6L80)

In clinical, the correlation between sympathetic nerve activation and plasma or urine levels is controversial. In contrast, the correlation between them was verified by various rodent stress models38-40. As described before, sympathetic hyperactivity following a single oral dose of FLs may occur as a stress response41. Therefore, we employed the determination of AD and ND levels in urine to estimate the change in SNS activity.

.

Discussion(p15L248)

Especially, the correlation between the activity of SNS and the level of plasma or urine CAs was verified in rodent stress models such as social defeat 38, hypothermia 39, or PTSD models 40. Oral administration of FLs likely activate SNS because they are considered stressor for the rodent.

Comment 6

Line 246~248, Does PRDM16 and PGC1a mediated BAT development through b3 receptor?

Response:We thank Reviewer 2’s comment. We added the statement as follows in P16L263.

These results suggest that the thermogenic formation of BAT through transcription of PRDM16 and PGC1α is possibly induced by repeated oral doses of FLs.

Comment 7

Figure 5b: Are the two pictures representative ones? Those may be too exaggerated ones?

Response:We carried out histological observation for iWAT taken from 5 of 11 mice in each group. We detected multilocular morphology in multiple sections (5-8 sections) in iWAT of FLs group, but there was no change in vehicle group. According to Reviewer 2’s comment, we added the statement as follows.

Results(p14L216)

In multiple sections in all of iWAT in FLs group, we observed the multilocular morphology along with a reduction in cell size, as shown in the right panel of Figure 5b. But these changes were not shown in vehicle group.

Comment 8

Line 277, there is no results showing that FL activated SNS, please edit this text in a prudent way.

Response:We revised as follows

Abstract(p3L38)

These results exert that FLs induce browning in adipose, and this change is possibly produced by the activation of the SNS.

Conclusion(p18L293)

In conclusion, the results of this study suggested that oral administration of FLs activated SNS, consequently induced the activation of BAT, and promoted browning of WAT.

Comment 9

Catecholamine downstream signaling pathway should be analyzed in BAT, eWAT, and iWAT. (PKA signaling, pHSL etc.)

Response:We thank Reviewer 2’s comment. We agree with Reviewer 2's opinion. It is important to further examination on the sympathetic nerve activating effect of FLs and the subsequent effect on adrenergic receptor-mediated adipose differentiation, and I would like to make this a topic for the future.

Minor points.

  1. The order of Figures should be fixed.

Response: We revised

  1. In the method section, most reagents were well described, however information for some reagents, for example secondary antibody for westernblot, are missing (company, city, country). Please review through the text

Response: We revised

  1. What is the meaning of upregulation of TMEM26, CD137 (pre brown adipocyte markers?) and PGC1a in eWAT but not in iWAT by FL?

Response: We thank Reviewer 2’s comment. In our experiment, when FL50 mg/kg was administered to male mice for two weeks, beige adipose was observed in iWAT. And gene expression of beige markers TMEM26, CD137 and heat production-related gene PGC1α was observed in eWAT. These differences were considered to be due to eWAT in male mice being resistant against SNS as shown in the Discussion. We think that browning of eWAT may be induced by longer-term or stronger FL stimulation.

  1. There are citation errors and text, for example references 30 does not have any data mentioned in the text

Response: We revised.

Round 2

Reviewer 1 Report

the authors have improved the pepar after the first revision 

Author Response

We are very grateful for Reviewer 1's decision.

Reviewer 2 Report

While authors tried to resolve some of the issues raised in previous reviews, the quality and quantity of data are primitive and explanation about their data are not clear enough. 

My specific concerns follow below,

Regarding:

Comment 2: 

Discussion (p15L238)

In addition, we observed that a single oral dose of FLs induced stress response with upregulation of corticotropin-releasing hormone (CRH) in paraventricular hypothalamic nucleus and increase of plasma corticosterone 41. The stressors result in rapid activation of the HPA axis and SNS, consequently, NA, AD, and corticosterone are secreted from the adrenal gland to the blood, while NE is secreted from the end of the sympathetic nerve to the effector including BAT 34. According to these previous results, it was suggested that oral administration of polyphenols activates SNS, resulting in thermogenesis through the β3 adrenergic receptor expressed in BAT.

As authors themselves acknowledged FL effect could potentially mediated not only B3 adrenergic, but also b1 and b2, this sentence should be edited unless provided more data.

Comment 3: 

Authors displayed figure without any mention in the text?

Is there morphological difference between control and experimental group.

What is the point of Figure 3a, 4a, and 5a?

Comment 4: 

I am not still convinced with their western blot and quantification shown in Figure 3b. Moreover, is statistics done properly?

Minor points 

1.: 

Figures 3, 4, and 5 are still ahead of Figure 2?

3. 

I don’t understand authors’ response about why there was no upregulation of beige markers in iWAT?

Author Response

Manuscript No.: Nutients-1401797

Title: Repeated oral administration of flavan-3-ols induces browning in mice adipose tissues through sympathetic nerve activation

Corresponding Author: Prof. Naomi Osakabe

Submit Date: 7 Nov 2021

RESPONSE TO REVIEWER 2: We wish to express our appreciation to Reviewer 2 for their insightful comments, which have helped us improve our paper significantly.

Comment 2

. (round 2)

As authors themselves acknowledged FL effect could potentially mediated not only B3 adrenergic, but also b1 and b2, this sentence should be edited unless provided more data.

Response:Thank you for the suggestion. We agree that more data is needed for understanding FLs action through SNS activation. While we already reported that FLs alter hemodynamics (PLoS One, 2014, 9, e94853) and this effect was exerted via β1 and β2 adrenaline receptors ((Free Radic Biol Med. 2016, 99, 584-592). We have now acknowledged this and added in the Discussion section of the revised manuscript as follows.

Discussion(p15L243)

According to these previous results, it was suggested that oral administration of FLs activates SNS, consequently, caused various physiological alterations. In our previous studies, a single oral administration of FLs increased blood flow in skeletal muscle 52 and this alteration may be exerted through the activation of β1 and β2 adrenaline receptors 53. Therefore, it was considered that FLs also induced thermogenesis through the β3 adrenergic receptor expressed in BAT.

(round 1)

It may be more interesting to understand how FL can be converted or FL stimulates catecholamine synthesis? If authors don’t have data, this should be speculated in the discussion.

Response:Therefore, increase of urinary CAs projecting  sympathetic activation may be induced by stress response by FLs . According to the comment of Reviewer 2, we added the statement as follows

Introduction (p5L70)

In addition, our previous results suggested that a single oral dose of FLs induced stress response. In the stress response, sympathetic hyperactivity occurs together with hypothalamic-pituitary-adrenal (HPA) axis activation34. Therefore, our previous results in these metabolic changes suggest that FLs possibly induce sympathetic nervous system (SNS) activation.

Discussion (p15L238)

In addition, we observed that a single oral dose of FLs induced stress response with upregulation of corticotropin-releasing hormone (CRH) in paraventricular hypothalamic nucleus and increase of plasma corticosterone 41. The stressors result in rapid activation of the HPA axis and SNS, consequently, NA, AD, and corticosterone are secreted from the adrenal gland to the blood, while NE is secreted from the end of the sympathetic nerve to the effector including BAT 34. According to these previous results, it was suggested that oral administration of polyphenols activates SNS, resulting in thermogenesis through the β3 adrenergic receptor expressed in BAT.

Comment 3

Authors displayed figure without any mention in the text?

Is there morphological difference between control and experimental group.

What is the point of Figure 3a, 4a, and 5a?

Response: Thank you for pointing out that our statement could be understood in this way. In Round 1, we asked Reviewer 2's if these photos only point to the excised parts and should be deleted. We regret that Reviewer 2 did not answer our question. Inferring from Reviewer 2s' context, we decided that it would be better to delete them, so we will delete these figures.

(round 1)

Are the pictures of Figure 3a, 4a, 5a tissues of FL treated? Where are the control tissues? And there is no description or explanation about these figures in the text at all.

Response:Photo 3a, 4a, 5a is just a photo of the excised part of each tissue. We will delete it if there is no particular need, but what should we do?

Comment 4

I am not still convinced with their western blot and quantification shown in Figure 3b. Moreover, is statistics done properly?

Response: We carried out western blotting as described in Method section (p10L27) and statistical was conducted (p12L188). We would like to collect data carefully in the future so that you will not be pointed out.

(round 1)

Although authors quantified the protein levels of UCP1, there seems to have very mild difference of UCP1 expression between control and FL applied tissues. Better to show more western blot results to convince the readers.

→We revised Figures 3,4,and 5.

Minor points.

  1. Figures 3, 4, and 5 are still ahead of Figure 2?

Response:Thank you for the suggestion. Since the experiment to measure urinary CA after a single dose of FL and the experiment to verify the effect of repeated doses of FL on fat were conducted in parallel, it seems that the order does not matter.

(round 1)

  1. The order of Figures should be fixed.

Response: We revised

  1. I don’t understand authors’ response about why there was no upregulation of beige markers in iWAT?

Response: In previous research that verified the expression of beige markers in mouse fat (10.1152/ajpendo.00023.2015), UCP-1 mRNA expression was increased in eWAT and iWAT of mice reared at 4 ° C for 30 days, but beige markers expressions (CD137, TMEM26, and TBX-1) have been reported to be reduced. In other words, it has been shown that when the increase in sympathetic nerve activity continues for a long period of time and the WAT completely differentiate into beige adipocytes, the expression of beige markers decreases and UCP-1 increases. Under the conditions of our study, the activation of sympathetic nerve activity was milder than 4 ° C for 30 days, so it was considered that iWAT was completely differentiated into beige, but eWAT was in the process of beige, therefore, expressing these markers.
